# A time of decline: An eco-anthropological and ethnohistorical investigation of mpox in the Central African Republic

**Romain Duda[1], José Martial Betoulet[2,3], Camille Besombes[4], Festus Mbrenga[5], Yanina Borzykh[1], Emmanuel Nakouné[5], Tamara Giles-Vernick[1]***

1 Anthropology & Ecology of Disease Emergence Unit, Department of Global Health, Institut Pasteur, Université Paris Cité, Paris, France, 2 Ndima Kali, Baaka and Sangha-Sangha Youth Association, Bayanga, Central African Republic, 3 Dzanga-Sangha Protected Areas (DSPA-WWF), Bayanga, Central African Republic, 4 Epidemiology of Emerging Diseases Unit, Department of Global Health, Institut Pasteur, Université Paris Cité, Paris, France, 5 Department of Virology, Institut Pasteur de Bangui, Bangui, Central African Republic

* tamara.giles-vernick@pasteur.fr

**Data Availability Statement:** Written requests for access to qualitative data must be addressed to the Institut Pasteur Institutional Review board,

## Abstract

The Central African Republic (CAR) has experienced repeated mpox outbreaks since 2001. Although several mpox epidemiological risk factors for zoonotic and interhuman transmission have been documented, the reasons for more frequent epidemic outbreaks are less well understood, relying on vague explanatory categories, including deforestation, hunting, and civil unrest. To gain insight into increasingly frequent outbreaks, we undertook an ethnohistorical, eco-anthropological analysis in two CAR regions: the Lobaye prefecture, experiencing one or more annual outbreaks in the past decade, and the Sangha-Mbaere prefecture, with a longer history of mpox but less frequent outbreaks. We comparatively examined changing political economies, forest use practices, and understandings of mpox. In 2022, we conducted 40 qualitative ethnohistorical, anthropological interviews and participant-observation of forest activities in two languages (Sango and French). We compared contemporary practices with hunting, trapping, and meet consumption practices, documented through quantitative and qualitative observation in one research site, over 6 months in 1993. We find increased rodent capture and consumption in both sites in the past 30 years and expanded practices of other potentially risky activities. Simultaneously, we also identify important differences in risky practices between our Lobaye and Sangha-Mbaere participants. In addition, Lobaye and Sangha participants underscored historical processes of decline producing mpox among other emergences, but they framed these declension processes diversely as economic, political, nutritional, and moral. Our findings are important because they mobilize new types of evidence to shed light on the processual dynamics of mpox outbreaks in the CAR. This study also reveals variability across two sites within the same country, highlighting the importance of comparative, fine-grained anthropological and historical research to identify underlying dynamics of mpox outbreaks. Finally, our study points to the need for mpox interventions and risk communication accounting for these regional differences, even within a single country.

irb@pasteur.fr. Written external requests for data access will be considered on a case-by-case basis. All data will be stored in .txt form (to ensure long-term data storage) on the Institut Pasteur internal drive for data storage. The Pasteur Drive is fully secured and regularly backed up. Should the IRB accept to share this data, encrypted data will be transferred to the Pasteur Cloud, where it can be shared with external individuals whose requests for data have been accepted.

**Funding:** This research was funded by the Agence Nationale de Recherche (Principal Investigator Arnaud Fontanet) (ANR 2019 CE-35). The funders had no role in study design, data collection and analysis, decision to publish, or preparation of the manuscript.

**Competing interests:** The authors have declared that no competing interests exist.

## Introduction

Even before its North American and European emergence in 2022, mpox (previously "monkeypox") [1] received heightened scientific attention in central and west Africa, where the virus has led to frequent outbreaks since its first identification in the 1970s [2]. Mpox, a virus in the Poxviridae family, can lead to mild disease or be fatal. Central Africa's Congo clade (clade I) often results in higher fatality rates than the pandemic West African clade (clade II) [3]. Epidemiological investigation in central Africa have documented increasing mpox case numbers since the 1990s [3,4]. Although poorly documented, these outbreaks purportedly commence with physical contact with wild animals and spread through close contact with patients in intrafamilial and health care settings. Although an animal reservoir has not been clearly identified [5], the tree-dwelling Thomas's Rope squirrel (*Funisciurus anerythrus*) is considered a probable candidate; other squirrels, giant-pouched rats, cane rats, striped mice, and dormice are also suspected, and multiple hosts may sustain the virus [6–10]. These rodent species are sources of protein throughout central Africa, and multiple reports find that wild meat handling, butchering, consumption, and hunting are implicated in mpox transmission [11,12]. A recent systematic review analyzing seasonal patterns of mpox cases in central Africa between 1970 and 2021 suggests that late rainy season/early dry season accounts for most cases in the northern equatorial forest [13]. Multiple studies have offered explanations for these outbreaks, including declining smallpox vaccine immunity [4,14], deforestation [11], hunting and contact with wild animals [5,15,16], and poverty [17]. The latter explanations are insufficient, however, because they rely on large categories, unsupported by concrete evidence of local human practices, local forest and animal ecologies, and political economic changes that shape viral exposures over time.

The Central African Republic (CAR) offers an illustrative site in which to examine in greater depth how and why recurrent outbreaks occur, given its diverse climatic and ecological profiles. Following its first documented mpox outbreak in 1984 in the Sangha-Mbaere region, the CAR has experienced multiple, small-scale outbreaks in the intervening decades [3,18,19]. National surveillance for this reportable disease was established in 2001, supported by the Institut Pasteur of Bangui (IPB), the CAR Ministry of Public Health and Population, and the World Health Organization (WHO). A recent analysis of national surveillance data between 2001 and 2021 showed 95 suspected outbreaks, of which 40 were confirmed, including 99 confirmed cases, 61 suspected cases, and 12 deaths, notably in the Lobaye and Mbomou prefectures [3]. A full epidemiological analysis identifying risk factors for transmission in CAR has never been done, although index cases reported contact with wild fauna (duikers, monkeys, rodents). Secondary intrafamilial and nosocomial transmissions have been documented, likely because family members and health care workers had close, unprotected physical contact with the sick person [20,21]. Since 2018, CAR has experienced more mpox outbreaks and cases, but for reasons not well understood. Although improved surveillance may account for some of the increase in identified mpox cases, the consistent trend suggests other unidentified factors at play. These cases have occurred in primarily rural settings, with some limited recent outbreaks in small towns (populations under 20,000) [3]. To date, in CAR and in central Africa more generally, we have identified neither anthropological nor historical analyses of local knowledge of recurrent mpox outbreaks, nor of livelihood practices potentially driving exposure to infected animals and facilitating these outbreaks. Such analyses could strengthen bottom-up public health prevention and response measures; their neglect has been a central criticism of One Health approaches that systematically sideline "nonscientific" knowledges [22–24]. Moreover, although civil unrest, deforestation, and heightened contact with wild animals figure prominently in scientific publications about mpox, such explanations offer little insight into

*why* outbreaks may occur more frequently in some sites than others, or the processes and specific practices that may increase risks of transmission and accelerate the frequency of such outbreaks in recent years.

To gain deeper insight into why mpox outbreaks may occur, we conducted an eco-anthropological, ethnohistorical investigation in two regions of the CAR. Our study had two aims: first, to gain insight into changing political ecologies of forest resource use, to shed light on past changes that can guide virologists, epidemiologists, ecologists, and global health actors in future research on mpox outbreaks in central Africa; and second, to analyze how local people perceive their changing lives and engagements with ecological, political economic, and social dynamics that can shed new light on recurrent mpox emergence [22,25].

## Materials and methods

As part of the multidisciplinary AFRIPOX project, our eco-anthropological, ethnohistorical investigation took place in the Lobaye (Mbaiki subprefecture) and Sangha-Mbaere (Bayanga subprefecture) of the CAR (Fig 1).

### Site histories and descriptions

These sites' histories reveal both commonalities and differences. The territory, now delineated by the boundaries of the CAR, has been a historical crossroads for centuries, and thus its

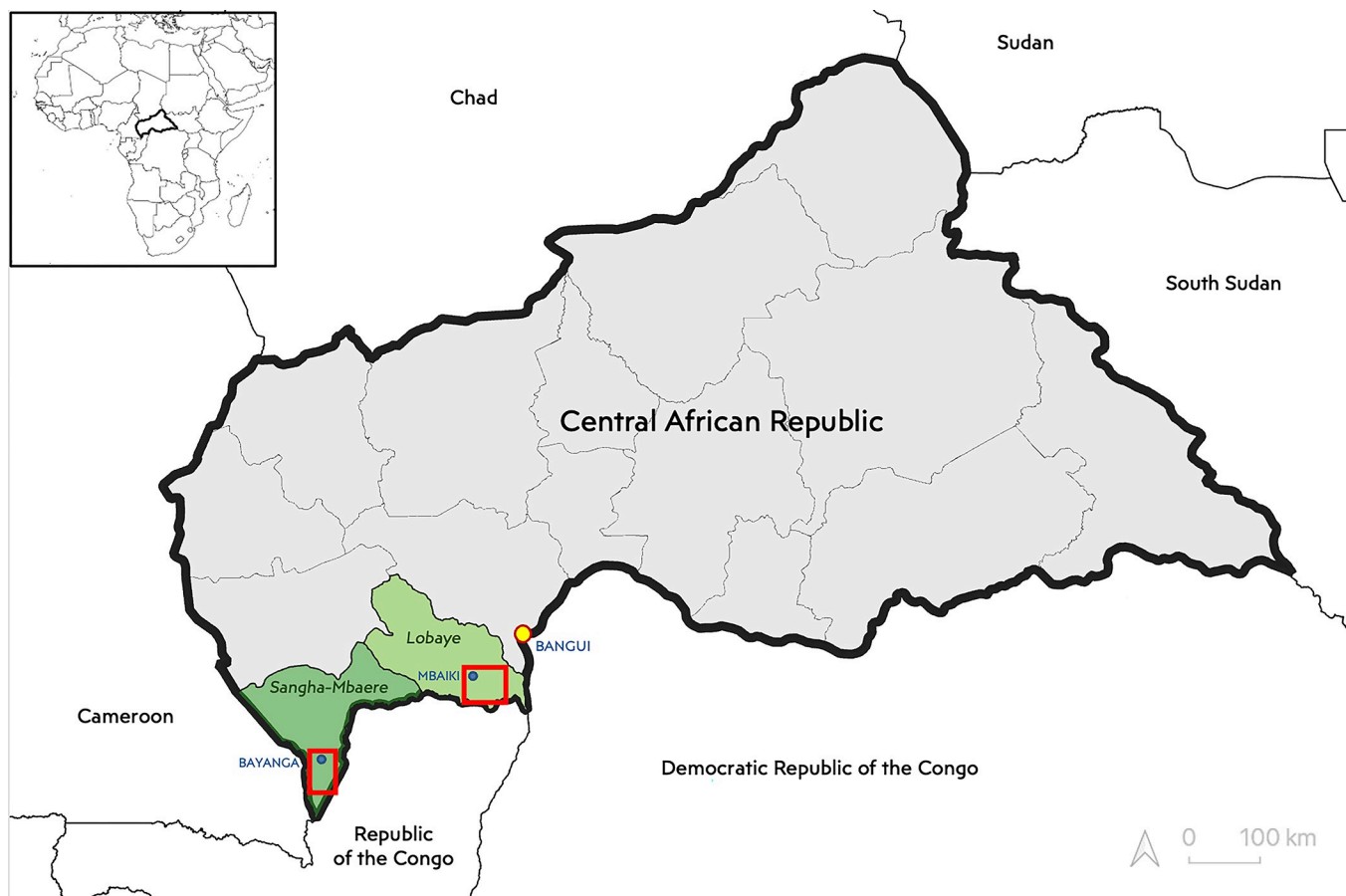

**Fig 1. Study sites are inside the red squares.** The map was created using QGIS, with boundary data from geoboundaries.org, provided under the CC BY 4.0 license. www.geoboundaries.org/countryDownloads.html / www.geoboundaries.org/globalDownloads.html.

populations have long experienced mobility, violent extractions, and political and economic upheaval [26,27]. Historical processes have differentially shaped this territory. During the 19th century, central Africa's integration into a global economy was characterized by slave raiding, high population mobility, intensive interaction between linguistic and ethnic groups, and violence, with the Lobaye region more closely integrated into Ubangi river and Bangui economic and political dynamics, and the Sangha region more implicated in trade and resource extraction along the Sangha river basin [28–30]. European colonization extended earlier processes of extraction, violence, and mobility, but differently. French colonial rule in Ubangi-Shari was established piecemeal over three decades from the late 19th century, initially through concessionary companies and eventually replaced by foreign-owned commercial enterprises [31]. The French colonial administration forcibly relocated scattered villages to designated roads, conscripted laborers to for colonial projects, and invested little in the colony's populations, economy, education, or health structures. Again, however, colonial rule differed in our two sites: the Sangha region, still integrated into a Sangha river network, experienced the rise and decline of coffee and cacao enterprises and trade in animal products, whereas the Lobaye remained dominated by extractive companies and was more closely linked to the colonial capital, Bangui. This lengthy history of extraction and violence throughout the CAR has led to a post-colonial period from 1959 characterized by high political instability and dependence on external political and economic support [27], including that of France, as well as regional differentiation and fragmentation. The CAR has experienced five coup d'états since independence, and from the early 21st century, the country has seen major instability, with warfare among multiple armed groups, economic crisis, and a tenuous governmental control over much of the country beyond the capital Bangui and southwestern parts of the country.

It is within this historical context of political upheaval, economic decline, mobility, and sedimented centuries of violence that repeated mpox outbreaks have occurred. Whereas both the Lobaye and Sangha-Mbaere prefectures have experienced mpox outbreaks, their epidemiological histories and the reach of government outbreak response differ. The Lobaye prefecture, CAR's most affected region, has seen increasing cases since 2018 [3], while the Sangha-Mbaere prefecture (site of the country's first documented and laboratory confirmed mpox case in 1984 [32]) has sustained far fewer and smaller outbreaks. Its distance from Bangui may render surveillance less consistent, possibly resulting in unreported cases. National surveillance (conducted collaboratively by the CAR Ministry of Public Health and Population, IPB, and WHO) for this reportable disease collects samples from suspected cases for laboratory analysis. There is no reporting threshold, so that all suspected cases are subject to surveillance and investigation. Health workers receive training to recognize clinical symptoms of mpox and to collect samples to send to IPB for laboratory analysis and confirmation. IPB also conducts mpox outbreak investigations during epidemics. Nonetheless, the reach of these surveillance, diagnostic activities, and outbreak investigations tends to be more extensive in the Lobaye prefecture than in the Sangha-Mbaere.

The Lobaye prefecture, 110 kilometers from the capital Bangui, is covered by IPB's frequent surveillance. Researchers from this institution regularly visit to collect samples, raise mpox awareness, and since 2021, treat mpox patients at the Mbaiki hospital [33]. Local economic activities include logging, small-scale artisanal gold and diamond mining, and primarily agricultural production and hunting. As elsewhere in the equatorial forest, inhabitants keep domesticated animals as a financial safety net, killing or selling their goats, chickens, or pigs when they need to pay for funerals, medical expenses, or school fees. Nonetheless, animal husbandry remains limited because of recurrent epizootics, including trypanosomiasis and Rift Valley Fever.

The Sangha-Mbaere prefecture, more isolated from large market centers, is situated in the Sangha Trinational forest (TNS), one of central Africa's largest wildlife conservation areas and

an UNESCO World Heritage site. This conservation area includes three national parks: Nou-balé-Ndoki (Congo), Lobeke (Cameroon), and Dzanga-Ndoki (CAR). Managed by the World-Wide Fund for Nature (WWF Germany), the Dzanga-Ndoki park permits no human activities except eco-tourism and scientific research; non-timber forest product (NTFP) gathering is permitted adjacent to the park, and hunting is authorized but regulated in the Community Hunting Zone.

In both sites, cultivation and forest resource extraction (fishing, hunting, gathering) varies across seasons and ethnolinguistic groups (including Bayaka, Ngbaka, Bangando, Gbaya, Sangha-Sangha, Mpiemu), who historically specialized in different subsistence activities [28–30]. Dry season activities include clearing new fields and fishing with different techniques. Hunting and gathering of forest resources, including caterpillars and mushrooms, is concentrated during the rainy season.

## Data collection

From October 23 to November 30, 2022, Romain Duda (a trained anthropologist with substantive research experience in central Africa) conducted ethnohistorical and anthropological research in Sango and French languages in selected villages of the Lobaye and Sangha-Mbaere prefectures. R. Duda was initially accompanied by Emmanuel Nakouné (virologist) and Tamara Giles-Vernick, a trained anthropologist and ethnohistorian with over three decades of ethnographic and historical research experience in CAR and central Africa and who conducted some interviews. In the Sangha region, José Martial Betoulet provided important assistance to R. Duda in contacting potential informants, providing Sango language translation, and contributing essential insights into local livelihoods.

Interviews were conducted with an interview guide, either individually or in groups of two to five people; occasionally, children or other adults would listen to the exchanges. Interviews lasted between 30 and 90 minutes. Researchers recorded all interviews following participant consent and took detailed notes. We did not seek saturation but sought to maximize the number of interviews for both sites.

We selected villages based on recent experience with mpox outbreaks. Recruitment was opportunistic; we began with village heads and local recommendations of knowledgeable informants, including elders, hunters, people previously affected by mpox and their family members. We approached potential informants in person, experiencing no refusals when we requested interviews. Conducted at participants' homes, interviews focused on changing ecologies and forest practices, broader political economic conditions, past and current diets, and local explanations for mpox and other new illnesses.

R. Duda also conducted participant-observation of multiple activities, including trap setting and rodent capture. During this participant-observation, he took detailed notes and filmed certain activities with a video camera.

In addition to conducting initial interviews in the Lobaye, T. Giles-Vernick had previously conducted extensive ethnohistorical, anthropological field research in the Sangha-Mbaere in 1993. This article also analyzes data from a small complementary study conducted during this 1993 research, specifically from a local meat market survey and a daily activity, income, and 48-hour dietary recall survey to compare wild meat consumption with that during our 2022 study. The activity, income, and 48-hour dietary recall survey took place between June and December 1993 among 24 adults (14 men, 10 women) between approximate ages of 20 and 85 living in a major village (Lindjombo) in the Sangha Mbaere prefecture. Convenience sampling was used, and recruitment procedures sought to include at least one member from each major extended family living in the village. Individual participants were included for their willingness

and availability to participate. Sample size was therefore not calculated. Because of very high mobility among village residents in 1993, participants were contacted in person periodically over the study period and asked to provide dietary, activity and income data. Data were therefore not collected each day for all 24 participants for the entirety of the six-month data collection period between June and December 1993.

### Data analysis

We used thematic analysis to evaluate our qualitative data [34,35]. We first transcribed our recorded interviews into French, although the transcriptions could not be returned to participants because of the unreliability of mail, nonexistent internet connections, and participants' inability to read. Following transcription, R. Duda and T. Giles-Vernick coded all qualitative data (transcripts and participant-observation notes) with Nvivo software (Release 1.7.1 (1534)). We conducted inductive coding of these qualitative data. Larger thematic codes included experience with mpox; terms for mpox; livelihoods; environmental changes in previous decades. The authors then consolidated codes into themes, which included local understanding of cutaneous diseases and mpox; understandings of health and associated practices; changing animal populations; changing access to resources, meat consumption patterns, and geographical mobilities.

Quantitative data concerning meat consumption from 1993 was entered into an excel file and descriptive statistics were performed using R software 4.1.1.

Participants did not provide feedback on our study findings because of their inaccessibility.

### Ethics

Ethical approvals for this study were provided by the University of Bangui (Comité Ethique et Scientifique, Université de Bangui, 21 February 2021) and the Institut Pasteur Institutional Review Board (IRB00006966, 10 January 2020). All participants received an information notice in French and Sango and a verbal explanation of their rights. They provided oral consent for participation in interviews, group discussions, and participation observation. We obtained oral consent because of the social sensitivity of mpox and of anthropological research in CAR. Potential participants might fear signing an informed consent form because of potential stigmatization of participating in a study on mpox. Both ethics committees approved this request. Consent was documented through each informant's participation in an interview, discussion, or participant-observation.

Ethical approval for the 1993 survey was not obtained because it was not required at that time for an ethnohistorical investigation.

## Results

Our qualitative interviews and participant-observation yielded several results concerning local conceptions of mpox, changing forest use practices, and ethnohistorical explanations for mpox outbreaks. Overall, we conducted 40 interviews with 29 men and 11 women (Table 1). The gender imbalance occurred because of a specific interest in hunting and trapping,

**Table 1. Profile of interview participants.**

| Location | Men | Women | Age range | Total |
|----------|-----|-------|-----------|-------|
| Lobaye | 12 | 5 | 32–73 (mean: 55) | 17 |
| Sangha | 17 | 6 | 28–78 (mean: 49) | 23 |
| Total | 29 | 11 | 28–78 (mean: 52) | 40 |

activities that are conducted by men and boys. R. Duda conducted 6 hours of participant-observation.

## Shared notions of mpox, but divergences between sites

Interview participants knew little or nothing about mpox and used multiple Sango and vernacular nosological terms that did not correspond directly to mpox. These terms reflected considerable fluidity in their use, shaped by various social and professional groups, patient age, and scale of the outbreak.

Informants in both sites defined illness based on symptoms, employing multiple terms for cutaneous illnesses that could include mpox. The Sango term for mpox is *yangba ti makako*, literally "cutaneous illness or pox of monkeys" and seems to have been popularized by Institut Pasteur of Bangui in their awareness-raising efforts. Nevertheless, this term inspired much confusion and debate among our informants. *Yangba* refers to a broad category of cutaneous illnesses, was historically used to denote smallpox [31], and in current use, connotes other cutaneous illnesses, including measles (*kete*–"small"–*yangba*) and chicken pox (*kota*–"big"–*yangba*).

In both regions, cutaneous diseases, including those with mpox symptoms, could be locally diagnosed variously, resulting from a *bricolage* of health worker assessments, healer-diviner influences, and family opinion. The age and numbers of afflicted people could also figure into the diagnostic term deployed. Among children, any cutaneous pox-like eruptions would more likely be explained as *kete yangba* (measles) or *kota yangba* (chicken pox). Because these illnesses occurred less frequently among adults, such adult afflictions were attributed to occult forces. Similarly, a single case resulted in a diagnosis of *yangba*, but several simultaneous cases, particularly among members of the same family, pointed to occult forces (and not to human-to-human transmission, as clinicians or biomedical researchers would suspect).

Mpox and its analogs are not generally understood as a zoonotic disease, although a few informants, influenced by the nosological term *yangba ti makako*, wondered if it resulted from contact with monkeys. Multiple informants, however, did not know what caused the illness, whereas several others believed that it was transmitted by wind (associated with other epidemic illnesses) or by drinking or stepping in contaminated water, which appeared to precipitate other cutaneous illnesses as well.

Recognition of mpox appeared more widespread among Lobaye informants than among Sangha-Mbaere ones. Several Lobaye informants contended that the illness was a new one and had come from the Republic of Congo, a consequence of recent intensified mobility between the two countries to supply meat and NTFPs to Bangui. Sangha-Mbaere informants more often invoked the risks of consuming wild meat, a message popularized by WWF and its radio station, Radio Njoku. These informants rapidly associated mpox, a new and largely unknown illness, with Ebola, a much-publicized illness but one for which there has never been an outbreak in the CAR. One hospitalized patient from Sangha-Mbaere revealed how rapidly and easily Ebola could be invoked:

> *"A few days before the ambulance came, people wouldn't go near me, even when I spoke to them. People would flee from me because they were thinking as they saw as my condition worsened that it wasn't measles, it was more likely Ebola. Even the doctor who came to see me was afraid and wouldn't approach me."* (Oral Account [hereafter OA].19, man, 45, Mbaiki hospital)

Such explanations of mpox cases as Ebola generated substantial fears of transmission and stigmatization of those suffering from the illness. Several informants noted that people

suspected of having mpox infection would flee, even before their admission to a health structure. Explanations for flight varied, including fears linked to local understandings of Ebola, blood collection, receiving care from medical staff wearing protective equipment, receiving an injection (understood to cause death among victims of occult forces), and dying alone in a hospital, widely perceived as a place to die.

### Livelihoods, hunting and exposures: Similarities and differences between Lobaye and Sangha

Our ethnohistorical, anthropological interviews also documented changing local socioeconomic, environmental, and seasonal practices that may have facilitated increased human-rodent physical contact and consumption. Alterations in hunting, trapping, and gathering practices, as well as changes in wild meat consumption have occurred in both study sites. These livelihood practices are important because they could bring inhabitants of the two sites into physical contact (through hunting, butchering, and meat preparation or handling for marketing) with an animal infected with the mpox virus.

In both Lobaye and Sangha-Mbaere sites, our informants consistently emphasized the lack of meat in their diets, although the histories and dynamics of access to wild meat differ across these sites. In the Lobaye, our informants contended that meat hunger, or a lack of wild meat, is not a new, but the result of long-term overhunting. As one informant observed,

"*in the 1980s and 90s, we already had strangers coming to hunt here. They came for the meat, trapping, and fishing. They evacuated all that they got, they brought firearms, bullets. . . They came from Mbaiki, Pissa, even Bangui.*" (OA.1, man, 65, Zoméa Kaka)

"Strangers" were those not living locally but moving between towns and the forest to supply wild meat to urban markets. Informants claimed that their forests were no longer home to large mammals and compared their fathers' prolific hunts with a current dearth:

"*We don't have any more monkeys here, we have to go very far now, even the dengbe [Philantomba monticola, blue duiker], even the genets, the mbengene [Cephalophus leucogaster, white-bellied duiker], ngbomu [Cephalophus dorsalis, bay duiker], there are no animals left. Today if we find the meat to eat then it is because it comes from the Congo.*" (OA.1, man, 65, Zoméa Kaka)

"*Before, my father used to set up traps and he found bemba, ngendi, mossome [three large duiker species], and dengbe [Philantomba monticola, blue duiker]. (. . .) Today we mainly eat dengbe, small animals, and monkeys, but sometimes they are rare. Our ancestors ate a lot of large animals!*" (OA.3, man, 70, Zoméa Kaka)

Informants noted that wildlife availability changed in the 1990s, when the Lobaye experienced a significant increase in low-cost shotguns from the Democratic Republic of Congo (DRC) and elsewhere in CAR. More recently, rising prices of bullets and of wire for snare traps have made hunting and trapping unpracticable for many village families. Informants also reported that their consumption of non-rodent wildlife was nowadays occasional, and that this absence increased reliance on rodents as sources of meat protein. Our interviews and participant-observations revealed that rodents are a highly democratized protein source, plentiful around settlements and cultivated fields, and requiring no specialized equipment. As one informant insisted,

"*In the past, people did not look for rats, there was enough meat*" (OA.4, village leader, man, 52 years old, Zoméa).

Whereas children frequently trapped forest mice and small rats, adolescents and adults tended to capture larger rodents (giant pouched rats and porcupines) using an array of techniques and knowledge (see video [36]). These generational practices were particularly salient among Bayaka peoples, who have historically been hunter-gatherers and who represent nearly half of the population in both studied areas. Participants generally reported greater success in trapping rodents during the early dry season (November-December), when they could use specific tree fruits as bait.

In the Sangha-Mbaere many informants similarly complained about inaccessibility of wild meat, although wild meat and hunting dynamics seem differ from the Lobaye. In that area, WWF and eco-guard patrols regulate hunting, and our informants there reported increased hunting pressures from a growing number of licensed shotgun hunters in the Community Hunting Zone (CHZ) and unequal access to meat because of declining game availability and unaffordable meat prices. Illegal snare trapping accrues heavy fines, but shotgun hunting is authorized with a permit, so that those who can pay for a shotgun, a license, and ammunition dominate hunting. In the early 1990s, Noss [37] recorded 8 legal and 12 illegal shotguns in Bayanga, whereas in 2022, some 60 hunters registered for firearm licenses there. Less well-off families, however, could not afford to hunt, and with the increasing scarcity of wildlife, found themselves unable to shoulder escalating wild meat prices in local markets (OA.5; OA.6; OA.8). As one Bayanga inhabitant noted:

"*today, we don't eat meat every day, where are we going to find the money to eat? They sell a ngendi haunch [Cephalophus callipygus] for 4000–4500 cfa [(6–7 €]! Where can you find this money?*" (OA.10, man, 56 Bayanga).

Although the high cost of hunting and meat and low meat consumption characterize contemporary conditions, our quantitative meat consumption survey, conducted by T. Giles-Vernick in 1993, provides a glimpse into wild meat consumption practices three decades ago. In that 1993 survey, we find that out of a total of 357 meal days recorded during that period in the same area, participants ate meat on 172 days. The most consumed animal was a large antelope, the bay duiker (*Cephalophus dorsalis*) accounting for 46,5% of all meat days (days during which participants consumed meat), followed by the red river hog and the giant forest hog (*Potomochoerus porcus, Hylochoerus meintertzhageni*), with 34,3% of these meat days. Blue duiker (*Philantomba monticola*) consumption was reported for just 2,3% of the days observed in 1993. No participants reported hunting, preparing, or consuming giant pouched rats (*Cricetomys emini*) or cane rats (*Thryonomys spp.*). By contrast, our 2022 interviews indicated that blue duikers were the most often consumed wild meat, that giant pouched rats and cane rats were frequent meat sources, and that large duikers and wild boars were rarely hunted. What is clear from our data collected in 1993 and in 2022 is that the size of wild animals consumed appears to have declined over this period. Moreover, the frequency of wild meat consumption also appears to have declined.

Our evidence also showed that in the Lobaye and Sangha sites, collecting certain NTFPs could also bring our informants into engagement with rodents. Seasonal edible caterpillar collection occurs between July and August in the Lobaye and in August in the Sangha-Mbaere. Informants maintained that their rodent trapping and consumption increased during caterpillar collection, because rodents are numerous (possibly attracted to human presence and food sources) and easily captured around forest camps by adults and children [36]. In the Sangha-

Mbaere prefecture, notably in Bayanga, palm wine extraction could also lead to heightened exposure to arboreal rodents. Palm wine is a fermented sap from various species of palm trees. Bayanga has extensive flooded forests of *Raphia vinifera*, from which collectors gather sap twice daily from receptacles placed in the upper reaches of the palm trunk. A recuperating mpox patient and daily palm wine collector from Bayanga noted that tree-dwelling rodents, such as squirrels and mice, sometimes drink from the palm wine receptacle at night:

> *"Squirrels come at night to drink out of the container, and sometimes they've drunk so much that they fall into it! Their abdomens expand, so they stay there. . .If we find a dead squirrel in the wine, we throw it away* (OA.13, man, 32, Mbaiki hospital)

He indicated that the wine would be kept and consumed. In Zoméa village (Lobaye), however, harvesters obtain palm wine from oil palms (*Elaeis guineensis*) by cutting down the tree to extract the sap. Harvesters reported that they did not find rodents around or in the sap.

## Local historical narratives address economic and moral decline, but diverge in their idioms and content

Local historical explanations focus less on *specific* explanations for mpox emergence than on new illnesses as a facet of larger, interconnected declines linked to social changes, economic recession, and problems of governance. Lobaye and Sangha-Mbaere women and men recounted declension narratives elucidating *why* repeated outbreaks of new illnesses occurred, employing different idioms and explanations.

Many Lobaye informants explained the decline in economy and health in terms of *blockages*. According to one 60-year-old woman,

> *"Before, we ate meat, fish, palm oil, but there isn't any anymore. Everything has changed. . .Especially from 2013, everything has gotten difficult, everything is blocked. People live badly: they have health problems; they have problems getting food. Before, there were roads, even paved. Taxis passed by, and even if you had a whole vat of manioc, you could get money for it. Now, the road is totally ruined, and the times have gotten really difficult. . . Safe drinking water? There isn't any. Same for health, and the hospital: no nurses, no pharmacists, no medicine."* (OA.12, woman, 60, Zoméa Kaka)

This quotation suggests three types of circulations that can be blocked: the forest itself, as a source of specific, valued resources (fish, caterpillars, animals); the pathways along which these resources can flow (roads, rivers); and the human body itself, resulting in illnesses that cannot be treated by an under-resourced (blocked) health system. Other informants contrasted prior circulations of forest resources and roads with current blockages, claiming that in earlier decades,

> *"The road here was open and usable, so you could sell resources directly in your village from people who came from Bangui. There were pickups, taxis, buses."* (OA.2, woman, 60, Zoméa Kaka)

Any village-wide epidemic, unpredictable rains, or a poor caterpillar harvest would thus be attributed to a blockage erected by someone employing occult forces to enrich themselves. Explanations of misfortune thus tended to be explained as the result of someone using occult forces, often motivated by personal gain. Accusations of sorcery, or the use of occult forces, frequently surfaced during situations of social tensions or rivalries within or between families.

Such accusations were regularly mobilized to explain illness or sudden deaths, particularly when these misfortunes simultaneously affected several members of a household who were previously in good health.

Many informants signaled that previously, these blockages were tackled collectively, although through different practices within the Lobaye's multi-ethnic populations. Bangando speakers employed a collective practice at the chief's house, by gathering leafy branches and shouting "Leave this country, leave this country, liberate us" to "chase away" the epidemic (*ewungo*).

Among Ngbaka speakers, *mogba* was a group of women or men using occult powers to "unblock" circulations essential to peace and wellbeing and to "block" the paths of those seeking to foment civil unrest and illness. A Ngbaka village head explained:

> "*A woman or a man who dances the mogba, we treat them as sorcerers. They dance naked, at night. You can't see them. Those who dance the mogba, they are sorcerers. But they can also cure. They make miracles, mystical acts. They can block people who want to install themselves in the village. They can do good things, like when the rebels wanted to enter the village during the war, the mogba people blocked them, and the road disappeared. They can simultaneously protect the village and destroy it. If there is an epidemic. . .that is also their role to block it.*"
> (OA.4, village leader, man, 52 years old, Zoméa)

For middle-aged and elder informants, *mogba* had ensured that forest resources flowed, that rain fell when it should, that roads remain unblocked to circulating traffic (but blocked to those seeking to inflict misfortune), and that illness outbreaks did not occur–or were displaced elsewhere.

Current blockages, however, were depicted as insurmountable, symptoms of a decline in a past political and moral order. Participants reproached their national political leaders, with one elderly man insisting,

> "*During the period of Dacko, of Bokassa [CAR presidents in the late 1960s to through 1970s], life was good. But from the time of Kolingba and Patassé [from the early 1980s and 1990s], things started to change. . .. [our sense of social solidarity] changed.*" (OA.3, man, 70, Zoméa Kaka)

Tellingly, David Dacko and Jean-Bedel Bokassa were from the Lobaye prefecture and of the same ethnic group (Ngbaka) as this speaker. André Kolinga, who initially came to power in a military coup in 1981, was also from southern CAR but of a different ethnic group. For this informant, his accession to power heralded for these informants the beginning of political and economic instability and their increased marginalization in national politics and access to financial resources from international political and economic interests.

During the same discussion, a younger man explained:

> "*The politicians have completely ruined everything. They have caused a lot of damage. This damage is felt in the lives of people here. There have been terrible things done to our lives, huge problems, maledictions to our bodies. . .We Central Africans, we've encountered some really heavy problems, and we started having those problems when our leaders didn't act well. It's then that we began to have these huge problems. . .*" (OA.3, man, 40, Zoméa Kaka).

Here, the younger man extended the reflections of the previous speaker about the political and economic marginalization experienced among Lobaye inhabitants at the hands of

politicians", or political elites. In referring to "Central Africans", he enlarged this marginalization to include a broader, non-elite central African population beyond specific ethnic groups or the Lobaye prefecture. Since the 1990s, CAR political strife has been characterized terms of regional (northern versus southern) origins, and for the first 15 years of the 21st century, religious affiliation. "Our leaders" referred to successive political elites from northern parts of the country whose access to wealth (including CAR's mineral, timber, and other resources) had been gained at the expense of non-elites.

On a local scale, several informants contended that the moral order that *mogba* had once defended had eroded: individuals in this group had been corrupted, seeking to accumulate money instead of supporting an entire village by facilitating flows of resources, money, and transport and preventing illness and instability (OA.3). Moreover, individual *nganga* (healer-diviners) now managed these blockages but commanded high payments for their efforts and were evidently less effective than prior collective efforts.

In the Sangha-Mbaere prefecture, we also found declension narratives describing why new illness outbreaks occurred. Like Lobaye informants, Sangha informants did not know where mpox came from and named an array of unknown illnesses that caused fear and precipitated flight from what they called "Ebola". New illness outbreaks were part of a recent history of economic and moral decline, but these narratives were more multifactorial and diffuse than those in the Lobaye region.

Sangha informants insisted that multiple dynamics precipitated this decline, notably the departure of foreign commercial enterprises which had supported economic activity and regular incomes, cleared forests to build and maintain roads, and sustained transport and sales of local goods. Some inhabitants still mourned the 1980s bankruptcy of a Slovenian coffee producer and the 1990s fall of state-supported coffee prices, spelling the end of coffee production and yearly income for coffee growing families. The bankruptcy of logging enterprises also deprived regional inhabitants of paid employment. The CAR state, rarely mentioned by informants, was a distant presence, disconnected from local priorities and needs. Conjugating with this economic decline were an increased reliance on the forest for meat, rising meat costs, and under-resourced health care structures. The upshot for local inhabitants was increased ill health and new illnesses. As one elderly woman elaborated,

*"Before, there weren't as many illnesses as there are now. Now, there are illnesses that we just don't understand. Lots of illnesses. . .have multiplied. . .and they kill people."* (OA.16, woman, 78, Bayanga)

A younger man, participating in the same interview, interjected:

*"I think that the illnesses we see are linked to the fact that people just don't eat well. Before, there was enough to eat. . . People ate three times a day. . . People now leave their homes in the morning and they don't eat until the evening. . . Your body will weaken, become more vulnerable to illness."* (OA.16, man, 45, Bayanga)

This man thus shifted the discussion from the unknown nature of and reasons for new illnesses to a more material explanation, suggesting that the poor nutritional status of regional inhabitants may have affected local vulnerability to disease.

Several informants also emphasized a moral decline, in which past knowledge about illness prevention was either disregarded by younger generations or no longer seen as effective. Many informants mentioned *egbale*, an older knowledge protecting previous generations from

misfortune brought by dead wild animals to hunters, trappers, and their families. As one middle-aged man explained,

> "*An animal cadaver, found in the forest, before, you wouldn't take it. You would avoid taking it, because you'd get egbale: all the children that you bring into the world would die. Now people have forgotten that, they have started finding dead animals in the forest, they take them, they eat them*." (OA.20, man, 33, Bayanga)

Some informants specified that only certain animals–notably rodents and pangolins— could confer *egbale*, and that even the act of seeing this dead animal could cause illness in an individual or entire family. As one informant explained, "*Egbale* depends on the animal. If you find a dead rat, without knowing the cause, as soon as you see it, you've taken on the *egbale*. And pangolins too, the small and large one" (OA.4, village leader, man, 52 years old, Zoméa). Another suggested that the abandonment of *egbale* knowledge stemmed from a desire to make money from selling dead animals found in the forest. (OA.16).

## Discussion

This study mobilizes new ethnohistorical and anthropological evidence to trace conceptions, changing hunting and meat consumption practices, and explanations for mpox among Central Africans directly affected by such outbreaks. We find that in two sites in CAR, lay populations had limited understanding of mpox and its symptoms and employed multiple, fluid nosological terms to refer to it. We also found reduced large game consumption and increased rodent capture and consumption in both sites, although our two sites were engaged in different economic and ecological dynamics that shaped local exposures to potential mpox infection. Finally, our informants explained the increase in new illnesses, including mpox, in terms of political, economic, and moral decline, although they framed their declension narratives differently.

Numerous articles have suggested multiple drivers producing increased mpox outbreaks: land use changes, forest clearing that expands the habitat for potential reservoirs, and climate change [5,38–40]. Ecological niche models have also shed light on mpox and candidate reservoir spatial distributions [6,38,41]. Although these studies identify large-scale drivers, they obscure specific practices and localized economic and historical changes bringing humans into contact with rodent reservoirs, and they fail to elucidate recurrent, varied mpox outbreaks, even in the same country. Drawing on Steven Hinchliffe's [42] rethinking of One Health, our ethnohistorical, anthropological approach framed mpox emergences in CAR as processual, localized events that emerged from "being entangled with others of various kinds", and that entailed long- and short-term changing relations in forest-animal-human-pathogen ecologies and in political economic and social relations.

### Mpox and local nosological terminologies

Although the Sango term for mpox is *yangba ti makako*, we found important regional differences in the uses of this term. The Lobaye's recurrent outbreaks, its proximity to Bangui, and frequent visits from Institut Pasteur of Bangui researchers clearly affected local population and health worker knowledge of *yangba ti makako*, the Sango term for mpox. In practice, lay populations readily linked mpox to other known skin afflictions. The proliferation of nosological terms and confusion that resulted whenever we asked questions about mpox, even when we described its distinctive pathognomonic symptoms, may result from multiple factors. For health care workers, mpox can be difficult to diagnose without virological confirmation, and

health workers may misdiagnose it as chickenpox [43]. Recent studies have also documented mpox co-infection with chickenpox. Such symptomatic resemblance and coinfection may understandably produce confusion [3,44], and local health workers may miss small-scale mpox outbreaks. Additionally, where lay diagnoses rely on perception of symptoms, lay publics can creatively expand nosological terms to refer to wide-ranging illnesses for which biomedical researchers and clinicians have distinct terms and etiologies [45–47].

*Yangba ti makako*, can also operate as a lexical field carrying specific meanings, so that some Sangha-Mbaere informants inextricably linked it with Ebola—not because of direct experience with Ebola or similarity in symptoms, but because of information circulating about the exceptional measures taken for patient isolation, evacuation, and treatment. They therefore responded with heightened fear to the arrival of health care teams dressed in protective equipment to evacuate mpox patients. We interpret this unexpected linkage that our informants made between mpox and Ebola in that region as an unintended and unpredictable consequence of WWF awareness campaigns for Ebola in previous years and for mpox in prior months. Highly implicated in forest management and more present that CAR state structures, WWF regularly produces radio programs about the risks of zoonotic emergence for people and protected gorilla populations there. That our Sangha-Mbaere informants interpreted signs of mpox infection as "Ebola" is of real local importance because it can affect how populations there react to further risk communication, especially in the event of an outbreak. Clearly, care is needed in framing mpox prevention messaging and risk communication. Because of the differences identified in how lay publics understand mpox in two prefectures in the same country, prevention messages and risk communication should be adapted to these distinct local understandings.

## Animal reservoirs: Insights from changing hunting offtakes and meat consumption

To understand better mpox animal reservoirs, several studies have linked cases to contact with wild animals, notably rodents, in days before infection [11,48,49]. Yet little is known about reasons for increasing primary transmission in specific sites. Our ethnohistorical, anthropological investigation opted for a more capacious contextual approach, exploring changes in species composition of hunting offtakes and meat consumption over the past 30 years. Our results highlighted increased exposure to rodents that may be infected with mpox virus [6,50]. In both sites, local populations depend heavily on wild meat for protein, although its availability and affordability differ, potentially leading to site-specific exposure patterns. Intensive hunting and heavy dependence on wild meat for consumption and income has deeper historical roots in extractive French colonial policy, initially through concessionary companies and subsequently through commercial enterprises [51].

Our Lobaye informants maintained that changing wildlife composition, political upheaval, and economic crises since 1980s have substantially increased reliance on rodents as a protein source. The Bangui and Mbaiki markets have exerted significant pressure on wildlife, leading to reductions in medium and large-sized mammals. Similar to a previous study in DRC [4], these wildlife pressures and political economic crises have compelled households to modify their prey choices, favoring animals that they consumed less in the past. In addition to much-appreciated giant pouched rats and brush-tailed porcupines, other rodents (e.g. mice, small rats and squirrels) are a reported alternative for adults, as shown elsewhere [52], likely because they are easily captured near human settlements, fields, and secondary forests, which are ideal habitats for rodents to sustain viral transmission [50]. These sites are also frequently used by children for playing and trapping. Aligned

with other human-rodent studies, we found that children and to a lesser extent, women also trapped rodents [48–51,53–57]. Boys from age five begin learning to hunt and trap small rodents and squirrels near their homes and in degraded forest and fields [36]. These activities could explain boys' higher mpox incidence [4,49,50,58]. More investigation of boys' rodent hunting practices is needed [59], particularly because mpox in children may lead to secondary transmission among caretaking adults [60].

Although mpox outbreaks may be under-reported in the Sangha-Mbaere prefecture, different hunting offtakes, constraints, and practices may also account for less frequent and smaller mpox outbreaks there. Our Sangha informants also reported declining medium and large-sized game populations over the past 30 years, but populations there appear less dependent on rodents than in the Lobaye. Its distance from market towns with high demands for wild meat may result in less hunting pressures. Other constraints on wild meat consumption, including WWF presence and the high cost of hunting permits, bullets, and meat, may also have moderated hunting pressures. At the same time, the disappearance of foreign capital, which previously drove economic activity, has forced inhabitants who can afford it to hunt (legally or not) in restricted areas, with an increase in numbers of hunters since 1994 [61]. Conservation policies and hunting regulations can produce imbalanced hunting pressures across fully and less protected areas and can possibly lead to different faunal composition and risky contacts even in the same forest zone.

The literature on faunal structure in African rainforests offers some insight into our results concerning changing hunting offtakes, declining wild meat consumption, and increased rodent consumption. Tropical forest faunal structures are not fixed, but a dynamic assemblage of living beings that interact with multiple ecosystem changes driven by climatic, agricultural, nutritional, demographic, political changes, and warfare [62,63]. Unsustainable hunting rates have been shown to alter species assemblages [64]. Ecological studies reveal that rodent predators tend to be preserved in protected areas, but that rodent abundance increases where their predators are hunted [65–67]. They tend to be an increasingly common prey in heavily hunted forests [68]. In the Lobaye, the disappearance of certain medium and large-sized species (e.g. red duikers) from hunting offtake indicates prior overhunting, so that inhabitants rely more on small, resilient and faster breeding mammals, including rodents and blue duikers [69,70], whereas Sangha faunal structures have not degraded so substantively. These ecological changes may explain a higher proportion of rodents in Lobaye diets, a region disproportionately affected by mpox.

Differences in the frequency and scale of Lobaye and Sangha-Mbaere mpox outbreaks might also result from other activities indirectly increasing human-rodent engagements. Although practiced in both regions, caterpillar harvest season is longer in the Lobaye than in the Sangha (possibly related to past selective logging practices that reduced host trees [37]). This longer harvest season may increase human presence in the forest and opportunities for adults and children to trap, prepare and consume rodents. Our results also suggest possible indirect contact with rodents through *Raphia vinifera* palm wine harvest in Bayanga area. Flooded swamps where this tree grows could be a human-squirrel contact zone. *Funisciurus anerythrus*, a species suspected in mpox transmission, is commonly found in agricultural areas, swamp forests and notably in palm trees [71]. Moreover, this setting may echo Nipah transmission studies in Bangladesh, where researchers found that bats alighted on, drank, and contaminated collected date palm sap, which was subsequently consumed by humans [72,73]. Explaining outbreak dynamics in these sites will require further multi-disciplinary investigation of changing faunal structures and hunting practices, as well as of caterpillar and *Raphia vinifera* harvesting.

## Local historical explanations for mpox and ill health

Finally, in both sites, local historical explanations attributed mpox and ill health to broader economic and moral decline. Although this result challenges biomedical expectations of a specific cause for a specific ailment, such broad ethnohistorical explanations do not seem to be unusual in central Africa. Here, wide-ranging factors elucidate broadly experienced declines, rather a specific disease emergence [25,46]. The historical processes to explain this focus on decline are multiple. As ethnohistorical researchers, we would underscore several longer-term processes that have subjected central Africans to political violence, extraction, and severe challenges in securing viable livelihoods and health care: central Africa's violent integration into a global economy beginning in the 19th century; highly extractive European colonial rule that invested little in local capacity; successive, volatile post-independence political regimes that depended heavily on French assistance (declining in recent decades, but still supporting the currency of the financial community of central Africa, which includes the CAR); and a global economy that has intermittently valued and devalued central African commodities [26].

More broadly, these narratives of decline raise crucial questions about global health security agendas, notably how, when, and for whom such agendas are constructed. Although CAR and other central and west African countries have experienced recurrent mpox epidemics in recent decades, mpox was not deemed a major priority for research or intervention until the 2022 global pandemic. In the words of one commentary, this neglect of mpox until 2022 reflected a collective failure in preparedness and "illustrate[d]. . .the double standard applied to how people's health is valued between wealthy countries and the rest of the world" [74]. It was only with intensified mpox transmission in wealthy countries that of mpox vaccines and treatment (tecovirimat) have been the focus of clinical trials and wider accessibility for people living on the frontlines of these recurrent emergences in central and west Africa. The narratives of decline recounted by our informants offer a powerful local commentary on this decades-long indifference to a recurrent health problem afflicting low-income countries.

For their part, our informants tended to highlight more temporally and geographically proximate economic, political, and societal declines from the 1980s. This period is characterized by reduced government support for all sectors because of International Monetary Fund-imposed structural adjustment programs, global economic shifts that reduced demand for central African-grown coffee and rendered global timber markets highly volatile. These explanations are valuable in themselves, offering "nonscientific" insights that illuminate how people at the frontlines of outbreaks understand linkages between new emergences and broader political economic and moral relations. Public health interventions to prevent mpox should consider these broader concerns.

We found greater coherence in declension narratives accounting for new illnesses like mpox in the Lobaye. New illnesses were indicators of blocked flows or circulations, reflecting accumulated experiences of prior decades of economic, political, and moral declines. Reproaches of political elites ("politicians") stemmed in part from a shared sense of political and economic marginalization among non-elites in the Lobaye. Our informants nostalgically recalled the early post-independence decades, when national political power, access to timber and mineral wealth and international capital were controlled by leaders from the Lobaye. But they seem to have understood the accelerating political and economic decline from the 1980s and 1990s in terms of a marginalization extending not just to Lobaye non-elites, but of a wider central African public. The coherence in expressions around present blockages and past circulations may have resulted from the Lobaye region's historical stabilization of multiple linguistic groups from the mid-nineteenth century [28]. In the Sangha, mpox emergence and overall poor health were more piecemeal and lacked a unifying metaphor like "blockages". Inhabitants

of this region have long had a sense of political marginalization from national political elites, and instead have seen their political and economic power subject to outside economic and conservation interests [29]. We hypothesize that these fragmented explanations may result from more recent histories of mobilities and a greater diversity of language groups in the Sangha river basin.

## Limitations of the study

This study has several limitations. First, we were unable to compare the two selected sites with others in the CAR that also experience recurrent mpox outbreaks because of insecurity. Thus, we cannot make claims concerning mpox in the entire country, and therefore limit our conclusions to two sites.

Second, our field research time was limited and would have been enriched by multiple field research trips during different seasons. We could not observe caterpillar harvesting or palm wine collection to identify potential risks. Nevertheless, this research benefited from the contributions and expertise of three central African researchers (J.M. Betoulet, Festus Mbrenga, E. Nakouné), from R. Duda's extended field research in Cameroon, Republic of Congo, and DRC (which included multiple observations of caterpillar collection), and T. Giles-Vernick's many years of field research (including palm wine harvesting) in CAR, Cameroon, and DRC.

Third, the 1993 survey of food consumption, activities, and income in Lindjombo, which provided data for our meat consumption comparisons, did not draw from a representative sample of the population, nor did it capture all participants' daily food consumption for the full six months during which T. Giles-Vernick collected this data. This survey was part of a different ethnohistorical study focusing on changing hunting, trapping, and other forest use practices. It offers, however, a rare indicator of changes in meat consumption over the past 30 years.

## Conclusion

This study, conducted in two regions of CAR, documented limited understandings and fluid nosological terms relating to mpox, reduced large game consumption, and increased rodent capture and consumption, as well as explanations for the emergence of new illnesses, including mpox, in terms of political, economic, and moral decline. We highlight the critical importance of attending to granular practices, regional differences, and ethnohistorical and social understandings of new emergences, but also of considering changes in multi-species interactions in the light of ecological transformations at very local scales.

Our investigation leads to several recommendations for mpox research and intervention. First, more multi-disciplinary studies of mpox that fully integrate anthropological, historical, and ecological insights are needed to illuminate localized, processual dynamics of emergence. Second, our study shows that training of health workers to manage mpox cases is crucial. That Lobaye health care workers could identify and respond quickly was a crucial advantage, in contrast to Sangha medical staff who displayed little familiarity with this disease. Finally, preventive measures, including risk communication about mpox, should account for local explanations linking mpox and ill health with broader political, economic, and moral declines. Such preventive measures should be developed collaboratively with local populations living on the frontlines of mpox outbreaks. These co-developed measures could include, for instance, locally-adapted health information about mpox that draws on local terminologies to reduce fears and to communicate specific precautions and steps to be taken by families. Above all, these co-developed measures should reflect their preoccupations, livelihoods, and variable practices around rodents to reduce mpox exposure risks without depriving them of a major protein source.

## Supporting information

**S1 Checklist. Inclusivity in global research.**
(DOCX)

## Acknowledgments

We sincerely thank inhabitants, health workers, and authorities in the Lobaye and Sangha-Mbaere prefectures for their warm welcome, openness, and friendship over the past decades. We are grateful to Benedetta Lana and Papa Mamadou Diagne for their valuable assistance in transcribing interviews and to Léonard Heyerdahl for his critical reading of the manuscript.

## Author Contributions

**Conceptualization:** Romain Duda, Emmanuel Nakouné, Tamara Giles-Vernick.

**Data curation:** Romain Duda, Tamara Giles-Vernick.

**Formal analysis:** Romain Duda, Yanina Borzykh, Tamara Giles-Vernick.

**Funding acquisition:** Emmanuel Nakouné.

**Investigation:** Romain Duda, José Martial Betoulet, Tamara Giles-Vernick.

**Methodology:** Romain Duda, Tamara Giles-Vernick.

**Resources:** Camille Besombes, Festus Mbrenga, Emmanuel Nakouné.

**Supervision:** Tamara Giles-Vernick.

**Validation:** José Martial Betoulet, Camille Besombes, Festus Mbrenga, Emmanuel Nakouné, Tamara Giles-Vernick.

**Writing – original draft:** Romain Duda, Tamara Giles-Vernick.

**Writing – review & editing:** Romain Duda, José Martial Betoulet, Camille Besombes, Festus Mbrenga, Yanina Borzykh, Emmanuel Nakouné, Tamara Giles-Vernick.

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
