## [Decision Letter · Decision Letter 0]

16 Oct 2023

PGPH-D-23-01319

A time of decline: an eco-anthropological and ethnohistorical investigation of mpox in the Central African Republic

Dear Dr. Giles-Vernick,

Thank you for submitting your manuscript to PLOS Global Public Health. After careful consideration, we feel that it has merit but does not fully meet PLOS Global Public Health’s publication criteria as it currently stands. Therefore, we invite you to submit a revised version of the manuscript that addresses the points raised during the review process.

We look forward to receiving your revised manuscript.

Kind regards,

Syed Shahid Abbas, MBBS, MPH, Ph.D.

Academic Editor

Journal Requirements:

2. We noticed that you used "unpublished data" in the manuscript. We do not allow these references, as the PLOS data access policy requires that all data be either published with the manuscript or made available in a publicly accessible database. Please amend the supplementary material to include the referenced data or remove the references.

Additional Editor Comments (if provided):

Thank you for sharing your manuscript to PLOS Global Public Health. Please consider clarifying the discussion on the lines suggested by Reviewer 2 & Reviewer 5.

Reviewers' comments:

Reviewer's Responses to Questions

**Comments to the Author**

1. Does this manuscript meet PLOS Global Public Health’s publication criteria? Is the manuscript technically sound, and do the data support the conclusions? The manuscript must describe methodologically and ethically rigorous research with conclusions that are appropriately drawn based on the data presented.

Reviewer #1: Yes

Reviewer #2: Yes

Reviewer #3: Partly

Reviewer #4: Partly

Reviewer #5: Yes

2. Has the statistical analysis been performed appropriately and rigorously?

Reviewer #1: N/A

Reviewer #2: N/A

Reviewer #3: N/A

Reviewer #4: I don't know

Reviewer #5: N/A

3. Have the authors made all data underlying the findings in their manuscript fully available (please refer to the Data Availability Statement at the start of the manuscript PDF file)?

Reviewer #1: Yes

Reviewer #2: No

Reviewer #3: Yes

Reviewer #4: No

Reviewer #5: No

4. Is the manuscript presented in an intelligible fashion and written in standard English?

Reviewer #1: Yes

Reviewer #2: Yes

Reviewer #3: Yes

Reviewer #4: No

Reviewer #5: Yes

5. Review Comments to the Author

Reviewer #1: Title:

It is important to have the full word Mankeypox instead of mpox => A time of decline: an eco-anthropological and ethnohistorical investigation of Mankey-pox (mpox) in the Central African Republic.

Line 6-11: It is good to number affiliation institutions from 1 to 4, so that is clear to understand the affiliation of each author.

Abstract:

Line 25: Is it really a mixed method or just qualitative study using different approaches. If this is the case, then delete the word mixed.

Line 29: … ethnohistorical, anthropological interviews in Sango and French and conducted participant-observation… If Sango is a language in CAR, then good to say “interviews in two languages: Sango and French. This helps even people who have been to CAR to understand the sentence.

Introduction:

Line 47: Need for a reference after endemic

Line 48: Need a reference after (clade II).

Line 64: The word “hereafter” is not necessary.

Line 68: Is the word “some” important in front of 95. Does it mean this number is not certain?

Line 85-87: The sentence “As part of the multidisciplinary AFRIPOX project, we conducted an eco-anthropological, 86 ethnohistorical investigation in two regions of the CAR with different frequencies and case numbers of 87 mpox outbreaks” should be part of methodology.

It is important to add information about risk factors of transmission between humans, humans-animals and local behavours. The authors say the disease is endemic in CAR, that means people have been exposed to it for long time. So, what is being done to respond to this public health issue at the community level and at the national level (health system)?

Materials and methods

Line 93: is it really a mixed method or just qualitative study.

Line 96 – 112: starting from the sentence: The territory now delineated … this part should be in the introduction.

From Line 132 in the whole method section: is it important to specify author 1, author 5, author 6,…? It is better to use general language as team to avoid frustration among authors.

Line 141: What does this sentence mean: …occasionally, children or other adults would listen to the exchanges? Were they part of the study participants or not? If no, why should they listen to the conversation? If yes, were children targeted for this study? And did it get the necessary approval for vulnerable group?

Line 142: … participants authorization… => I hope this means participant consent. Use the commonly known language. Was the consent verbal or written? Did they sign consent form? Please specify if participation was voluntary or paid.

Line 143: We did not seek saturation but sought to maximize the number of interviews for both sites. => How many participants per sites and how the participants were selected (sampling method)?

Line 150-152: Please specify here what you mean by authorization and oral consent. Use the commonly known language for ethics.

Line 155-159: The word survey just appear suddenly in the middle of the text. This means the study was really mixed methods. However, it is not clear at the beginning. This should be stated everywhere it is said mixed method and in the abstract. How did the authors select participants for the survey (sampling method)? And why 24 participants? Is this number sufficient for a survey? Was there a sample size calculation? How?

Line 157: ” Because of very high mobility among village residents at the time,…” what time?

Line 157-159: “Because of very high mobility among village residents at the time, participants were contacted in person periodically over the study period to provide dietary, activity and income data; data was therefore not collected for all 24 participants for the entirety of the six-month data collection period.” Please resolve issue of grammar in this sentence and you can slip it into 2.

Line 160: It is better to start this paragraph by explaining that you used thematic analysis then continue explaining how as written here. Was there any conceptual framework used?

Lines 160-169: It is not clear how the thematic analysis was conducted. Inductive or deductive method? It is also not clear which data were anaylised with Nvivo (qualitative or survey) and which ones were analysed using R software (qualitative or survey)?

Ethics

Line 171-178: If I understand well, written consent was not suggested because of safety to avoid mpox transmission. If that is the case, then it should be clearly stated here, so the readers do not have to assume. This section does not say anything about consent for survey questionnaire. Was it not needed? Why?

Information about ethics were stated many time in the previous section in the method. It is important to separate them and put everything about ethics in this section (171-178).

Results

Line 180-186: This paragraph should be part of the methods

Line 212: Is this “Republic of Congo or Democratic Republic of Congo”? They are two different countries. Be sure which you refer to here.

Line 218-221: Is this a quote? If yes, it should have quotation mark “….”

Line 234: What do authors mean by “meat hunger”?

Line 235-237: This should be in a quote format.

Line 275-281: Are these data reported here part of this study? If yes, it should be stated clearly from the beginning and in the methods. If no, is there an unpublished report that the author refer to? It could be better to have this information in the introduction as part of the literature to support the rationale.

Line 295: Add quote mark at the beginning.

Line 308-315: Put quote mark.

Line 332-339: Put quote mark.

Line 349-354: Put quote mark.

Line 356-358: If this is a quote, then show it.

Line 375: …Lots of illnesses… should read Lot of illnesses…

Line 378-381: Please put quote marks here. And all the quotes should have the same format.

Line 389-392: Please put quote mark.

Discussion

Line 400-408: This paragraph has a mix of many ideas. It would better to separate paragraphs by ideas (themes).

Line 430-431: Difficult to understand why sleeping sickness is referenced here.

Line 434-442: Can the message here be made clear? It can be interpreted that WWF made a mistake in its awareness campaign.

Line 490: Is the word “see” relevant here?

Line 498: The sentence “…further multi-disciplinary investigation…” should read “…further multi-disciplinary investigations…”

Line 528-529: “This unpublished data was collected for a different study. It offers, however, an indicator of changes in meat consumption over the past 30 years.” => fix issue of grammar in this sentence.

Conclusion

Line 545: Good idea of co-development of solutions with local populations. However, this come only in the conclusion. Nothing is seen in the results and rationale about it.

Line 547: Why reference in the conclusion?

Line 549-554: Acknowledgement should at the very beginning before abstract.

Line 556: Is this Bibliography or just a list of references?

General comments:

- Fix the issue of page number formatting

- All the titles are too small and do not comply with PLOS GPH title guideline.

- The manuscript is scientifically relevant and carry the message that can inform policy and programs to improve public health interventions. Following the above review, the corrections are important to be made before further considerations.

Reviewer #2: This mixed methods study used ethnohistorical and eco-anthropological analyses based on ethnographic work in two CAR regions to understand the political economies of mpox, and underscores the continued importance of anthropological work in the space of outbreaks and outbreak response. It is an excellent and important piece of work which could be further strengthened with some substantive revisions.

Firstly - I think it’s very difficult to speak about political economy without mentioning colonial history, particularly in a former French colony (and continued French intervention). What role might this history play in the wider political economy of mpox?

Consider spending a little more time discussing dynamics related to human-to-human transmission, as mentioned in the introduction. Thinking of other mpox-endemic areas, like Nigeria, has there been similar spread from rural areas to urban? Perhaps something to signpost, not necessarily go in-depth here, but do any of your findings speak to this?

Some minor comments:

- This is a very well-written article, but please check some of the English language words as the meaning may be lost or misinterpreted, for example:

o e.g., page 4 – line 55, ‘manipulation’ – does this mean butchering?)

o page 4, line 61 – instead of ‘weakened,’ consider ‘insufficient’

- Page 5, line 69 – any deaths noted?

- Page 6 – can you add details about the colonial history and how this likely laid the groundwork for the dynamics (conflict, etc.) that you describe? I am sure that French colonialism played a big role in these current dynamics, please add detail here, otherwise it reads quite negative of the CAR.

- Page 8, Line 142 – participant ‘consent’ instead of authorization?

- Page 8, line 146 – were individuals suspected or lab confirmed mpox cases?

- Page 8, line 163 – do you mean ‘thematic’ codes, rather than large codes?

- Page 8, Line 165 – do you mean skin diseases, not cutaneous?

- Line 213 – what is NTFPs? Please spell out all acronyms the first time it’s mentioned.

- Line 234 – what is meat hunger?

In terms of the section on livelihoods, hunting, etc. – you may want to add more to the first paragraph about why this section relates to mpox transmission (possibly).

The section on local historical narratives and ‘blockages’ is very interesting, but perhaps we need more background about cultural context. In what situations might occultism be used?

Also, I would like to see more explicit attention to power and power inequalities within your political-economic analysis. What or who do people blame for the general state of decline? You mentioned the politicians, but what are these ethnic groups’ experiences of mainstream politics, are they a politically or socially marginalized group?

One other thing to consider, perhaps in the discussion. How do your findings problematize or shed light on the global health security agenda, and how/when certain diseases are declared a ‘public health emergency’? How do these heterogeneous experiences, particularly in endemic regions (vs the global multicountry outbreak), shape when and how something is declared an emergency? And then how does this shape response, funding flows, etc.? Some reflections on these points could add strength to the global importance of the work.

Reviewer #3: Thank you for the opportunity to review the paper. It is a very interesting read, and gave different perspective to the mpox emergence in one of the most challenging context, the Central African Republic. It is of note that the paper dislodge the biomedical superiority as the only lens used to understand and control a ‘disease’, acknowledging the people’s story and narrative of what afflicted them. Hence I commend the team to spend time in this important piece of evidence and further increase awareness of much needed multidisciplinary approach to a complex problem.

The study is an investigation conducted in 2 regions in CAR with different endemicity and outbreak proneness related to mpox. Data was collected through interviews, participant observations and comparison with data from 30 years ago.

Certainly, it is a long read and I wonder how much has been cut as well as not to bore the reader but keep them engaged. This is a type of study that requires the reader to step out of the strict qualitative research methodologies and appreciate the richness of information.

My suggestions to the research team to improve the manuscript are as follows:

- Line 25: the term mixed method -does this refer to quantitative and qualitative assessments or in just to say different methods were deployed?

- Line 30-31: a database of historical data is mentioned, but with one time point only i.e 30 years ago; is there no data from the time in between 1993 and 2022?

- Line 33-34: The sentence ‘We also find important differences..’ belongs closer to Line 38 (.. reveals variability..) and not the similarity of the two sites in terms of their declension narrative

- In the introduction, line 66 onwards – what is the ‘outbreak’ term refers here i.e what is the threshold of declaring it as outbreak in CAR? Is it a notifiable disease that must be reported to the health authority? Is one case enough to declare so?

- Reference given in that paragraph is duplicate: reference 3 and 19 is the same

- Line 102: The CAR – suggest removing the word ‘The’

- Line 117-119: these 3 last sentences is unclear to read, suggest rephrasing

- On material and methods:

- Given the high mobility of the population, and the diversity of the tribes, how did selection bias be mitigated?

- What was the reason not to seek saturation during the data collection especially with sampling strategy of opportunistic participant recruitments and snowballing (asking key informants)

- How do you differentiate the reporting of individual versus group interviews or both are combined in Table 1? Why not doing focus group discussion?

- Participant observation time seems quite short, would there be any reasons for this?

- I appreciate clarity on the positionality and reflexivity of the research team

- In results, the quotes are organized sometimes integrated in paragraph, sometimes not, please standardized (example: line 235,, 258, 345..)

- Line 261 – is supplemental file 1 a text documenting rodent trapping techniques? (supplemental file I see is the video if not wrong)

- Line 274 beyond – the description of eating meat habits may be more specific? Meat days means once per day? Per person or per family? Is it the size change of the animals hunted in 1993 the main thing we need to understand?

- Line 301 and elsewhere: is the word ‘moral’ used as ethical understanding that relates to religion or beliefs in this population?

- The discussion is well written but the implications can be theorized – e.g would paragraph starting line 419, would mean the messages at least to seek care should consider the nosological terms? Or would health workers be more aware and alert on the description of symptoms and also for notification of cases?

- Please clarify if the Ebola messaging have influenced the way mpox is perceived or experienced being not lethal disease?

- The discussion is well written, albeit could benefit

It is an interesting read and definitely help us appreciate the complexity of such disease ‘emergence’. Not the easiest read as well so I hope the feedback above can improve

Reviewer #4: I want to thank the authors for putting together this insightful manuscript on “A time of decline: an eco-anthropological and ethnohistorical investigation of mpox in the Central African Republic”.Authors seek to explain why outbreaks may occur more frequently in some sites than others, or the processes and specific practices that may increase risks of transmission and accelerate the frequency of such outbreaks in recent years. This is done through anthropological and historical analyses of local knowledge of recurrent mpox outbreak and livelihood practices in CAR. However, needs major structural revision which will affect the scientific soundness.

Abstract:

The abstract as its current state is difficult to comprehend. I suggest a structured format of writing an abstract be used. This will help readers better appreciate the content.

Introduction

Line 74, how has improved surveillance led to an increase and of what specifically?

Generally, epidemiological data on mpox outbreaks are missing. This data will be important to understand the ethnohistorical situation within CAR.

Materials and Methods

The structure is poor. No information on how data was managed, study period etc. Authors need to clearly document all steps using sub-headings.

Results

Can we have subheadings for all results presented?

Discussion.

This section is well written and backed by science but difficult to make judgement as to the soundness because other previous sections such as the method are not still vague.

Generally, the article will contribute greatly to global health security and prompt reproducibility for more findings that can help in the fight against zoonotic diseases.

Reviewer #5: Broadly speaking, this article is an interesting read that fits with the aims and scope of the journal and nicely demonstrates the value of ethnographic research in a public health context. I look forward to being able to assign it to students in relevant courses.

My recommendations for revisions are relatively minor and mainly focus on clarifying the data, methods, and conclusions. First, I recommend including both a map indicating the study regions as well as Bangui and a table summarizing similarities and differences relevant to the analysis of the two study sites. Similarly, a figure of some kind in the discussion, such as a flowchart, showing broader conclusions of how different factors are related to each other and to health outcomes could be useful.

Second, there are only minor and insufficient references to the supplemental file video. It is unclear what the purpose of including this video is and what readers should take from it, i.e., what it adds that the main text is unable to address.

Third, including details of the 1993 field research and related data with the rest of the data collection in the methods caused confusion for me. Beyond wondering how it was applicable to the current study and trying follow what data was from which collection period, I also then anticipated much more use of it in the later analysis. Personally, I think it would be fine to just briefly describe it the first time it is brought up in analysis/discussion, as one would with cited previous research, rather than highlight it with the main data collection.

Finally, I would like to see some additional discussion, if possible, of: a) differences in perspectives based on age, b) how the study could be used to improve public health communication and interventions. For example, what might a collaboratively developed preventive measure “look like”?

On a minor point, there were a number of small grammatical errors, mostly missing or extra words, so be sure to review thoroughly. I do not include page numbers to indicate specific sections because the numbering was all messed up on my copy.

6. PLOS authors have the option to publish the peer review history of their article (what does this mean?). If published, this will include your full peer review and any attached files.

**Do you want your identity to be public for this peer review?** For information about this choice, including consent withdrawal, please see our Privacy Policy.

Reviewer #1: No

Reviewer #2: No

Reviewer #3: No

Reviewer #4: No

Reviewer #5: No

---

## [Decision Letter · Decision Letter 1]

18 Jan 2024

PGPH-D-23-01319R1

A time of decline: an eco-anthropological and ethnohistorical investigation of mpox in the Central African Republic

Dear Dr. Giles-Vernick,

Thank you for submitting your manuscript to PLOS Global Public Health. After careful consideration, we feel that it has merit but does not fully meet PLOS Global Public Health’s publication criteria as it currently stands. Therefore, we invite you to submit a revised version of the manuscript that addresses the points raised during the review process.

Thank you for considering the reviewer comments in revising your manuscript. The reviewers all agree that the manuscript is substantially strengthened as a result of these revisions. I look forward to seeing this manuscript published after you have addressed the remaining comments which are mostly clarifying in nature. 

We look forward to receiving your revised manuscript.

Kind regards,

Syed Shahid Abbas, MBBS, MPH, Ph.D.

Academic Editor

Journal Requirements:

Additional Editor Comments (if provided):

Reviewers' comments:

Reviewer's Responses to Questions

**Comments to the Author**

1. If the authors have adequately addressed your comments raised in a previous round of review and you feel that this manuscript is now acceptable for publication, you may indicate that here to bypass the “Comments to the Author” section, enter your conflict of interest statement in the “Confidential to Editor” section, and submit your "Accept" recommendation.

Reviewer #1: (No Response)

Reviewer #2: All comments have been addressed

Reviewer #4: All comments have been addressed

Reviewer #5: All comments have been addressed

2. Does this manuscript meet PLOS Global Public Health’s publication criteria? Is the manuscript technically sound, and do the data support the conclusions? The manuscript must describe methodologically and ethically rigorous research with conclusions that are appropriately drawn based on the data presented.

Reviewer #1: Yes

Reviewer #2: Yes

Reviewer #4: Yes

Reviewer #5: Yes

3. Has the statistical analysis been performed appropriately and rigorously?

Reviewer #1: N/A

Reviewer #2: N/A

Reviewer #4: Yes

Reviewer #5: N/A

4. Have the authors made all data underlying the findings in their manuscript fully available (please refer to the Data Availability Statement at the start of the manuscript PDF file)?

Reviewer #1: Yes

Reviewer #2: No

Reviewer #4: (No Response)

Reviewer #5: No

5. Is the manuscript presented in an intelligible fashion and written in standard English?

Reviewer #1: Yes

Reviewer #2: Yes

Reviewer #4: Yes

Reviewer #5: Yes

6. Review Comments to the Author

Reviewer #1: Title:

A time of decline: an eco-anthropological and ethnohistorical investigation of mpox in the Central African Republic.

Abstract:

OK

Introduction:

Line 69: … 99 confirmed … => I think this should be 99 confirmed cases.

Materials and methods

From Line 186-187: The following sentence is not clear: “Because of very high mobility among village residents in 1993, participants were contacted in person periodically over the study period (between June and December 1993)”. What is the study period (1993)? This is confusing because data were collected between 23 October to 30 November 2022.

Line 188-189: I don’t understand where this sentence is coming from: “Data were therefore not collected for all 24 participants for the entirety of the six months data collection period”. It does not have connection with the previous part. Why 24 participants? Where is this figure coming from?

Results

Line 268: Can authors give details about the socio-economic changes? What have changed socially or economically?

Discussion

OK

Conclusion

OK

General comments:

- Fix the issue of page number formatting

- The manuscript is scientifically relevant and carry the message that can inform policy and programs to improve public health interventions. Following the above review, the corrections are important to be made before further considerations.

Reviewer #2: (No Response)

Reviewer #4: Thank you once again for this write-up. I enjoyed reading it and reviewing it. I belief it will contribute immensely to global health security. The findings should be made available to the public as much as possible.

Reviewer #5: (No Response)

7. PLOS authors have the option to publish the peer review history of their article (what does this mean?). If published, this will include your full peer review and any attached files.

**Do you want your identity to be public for this peer review?** For information about this choice, including consent withdrawal, please see our Privacy Policy.

Reviewer #1: No

Reviewer #2: No

Reviewer #4: No

Reviewer #5: No

---

## [Editor Report · Decision Letter 2]

31 Jan 2024

A time of decline: an eco-anthropological and ethnohistorical investigation of mpox in the Central African Republic

PGPH-D-23-01319R2

Dear Dr Giles-Vernick,

We are pleased to inform you that your manuscript 'A time of decline: an eco-anthropological and ethnohistorical investigation of mpox in the Central African Republic' has been provisionally accepted for publication in PLOS Global Public Health.

Best regards,

Syed Shahid Abbas, MBBS, MPH, Ph.D.

Academic Editor